# Origin of breath isoprene in humans is revealed via multi-omic investigations

Pritam Sukul [1,3 ✉], Anna Richter [2,3], Christian Junghanss[2], Jochen K. Schubert[1] & Wolfram Miekisch [1]

Plants, animals and humans metabolically produce volatile isoprene ($C_5H_8$). Humans continuously exhale isoprene and exhaled concentrations differ under various physio-metabolic and pathophysiological conditions. Yet unknown metabolic origin hinders isoprene to reach clinical practice as a biomarker. Screening 2000 individuals from consecutive mass-spectrometric studies, we herein identify five healthy German adults without exhaled isoprene. Whole exome sequencing in these adults reveals only one shared homozygous (European prevalence: <1%) *IDI2* stop-gain mutation, which causes losses of enzyme active site and $Mg^{2+}$–cofactor binding sites. Consequently, the conversion of isopentenyl diphosphate to dimethylallyl diphosphate (DMAPP) as part of the cholesterol metabolism is prevented in these adults. Targeted sequencing depicts that the *IDI2* rs1044261 variant (p.Trp144Stop) is heterozygous in isoprene deficient blood-relatives and absent in unrelated isoprene normal adults. Wild-type *IDI1* and cholesterol metabolism related serological parameters are normal in all adults. *IDI2* determines isoprene production as only DMAPP sources isoprene and unlike plants, humans lack *isoprene synthase* and its enzyme homologue. Human *IDI2* is expressed only in skeletal-myocellular peroxisomes and instant spikes in isoprene exhalation during muscle activity underpins its origin from muscular lipolytic cholesterol metabolism. Our findings translate isoprene as a clinically interpretable breath biomarker towards potential applications in human medicine.

[1] Rostock Medical Breath Research Analytics and Technologies (ROMBAT), Dept. of Anesthesiology, Intensive Care Medicine and Pain Therapy, University Medicine Rostock, Schillingallee 35, 18057 Rostock, Germany. [2] Department of Medicine, Clinic III – Hematology, Oncology, Palliative Medicine, Rostock University Medical Center, Ernst-Heydemann-Strasse 6, 18057 Rostock, Germany. [3]These authors contributed equally: Pritam Sukul, Anna Richter. ✉email: pritam.sukul@uni-rostock.de

soprene (2-methyl 1,3-butadiene; $C_5H_8$) is an omnipresent and the most abundant hemiterpene in our planet's atmosphere[1,2], Isoprene is globally produced by vegetation and along with its chiral monoterpenes (e.g., alpha pinene) acts as predictor for ecosystem fluxes[3], forest emissions and drought response[4], environmental pollution[5], cloud chemistry[6], and climate change[7]. Besides being abundantly emitted via the de novo emission pathway in plants[8], isoprene is also endogenously produced by animals and humans.

The $C_5$ isoprene unit is the basic building block for terpenoid including steroid hormones biosynthesis that has been biologically essential for ubiquitous terrestrial life forms since billions of years in the course of evolution[9]. It is the second most abundant and endogenous volatile organic compound (VOC) in our exhaled breath[10]. Exhaled concentrations range between 80 and 300 ppbV in healthy adults. In 1960, mass-spectrometric techniques had detected the presence of exhaled isoprene, which was later quantified by Jansson et al.[11,12] Though the evolutionary significance of isoprene production and biological function are not well understood in humans. differences/dynamics in its exhaled alveolar concentrations are well reported as potential markers for various physiological, metabolic, and pathophysiological effects. While any kind of muscle activity/exercise immediately spikes its expression[13,14], studies have reported differential breath profiles and significant concentration changes under conditions such as cardio-respiratory diseases[15–17], hypercholesteremia[18], oxidative stress[19], cancers[20,21], sexual arousal[22], and ageing[23–25]. Despite such interesting observations, the clinical translation of isoprene as a routine biomarker is hindered due to the uncertainty upon its exact metabolic origin.

Surprisingly, while significant deficiency or absence of nonsterol $C_5H_8$ is reported in cases with inherited diseases like mevalonic aciduria, hyper immunoglobulinemia D syndrome, autoinflammatory periodic fever[26–28] and Duchenne muscle dystrophy[29], there are rare (<0.3% population prevalence) adults living healthy lives without (absence) exhaling any traceable isoprene and exhaling very low (deficiency) concentrations of isoprene[30]. Existence of an isoprene absent adult was first reported by Gelmont et al.[31] Since then, only a few pilot studies have randomly reported the presence of breath isoprene absent healthy subjects and in 2021, we have first approximated the actual rare genetic occurrence of this character by screening in a large cohort of 1026 humans[30]. Such breath isoprene aberrated healthy adults may hold the fundamental key to its true origin in human breath[30].

While looking at the physio-metabolic aspects, the increase in isoprene exhalation at the beginning of exercise was first reported by Jordan et al.[32] Miekisch et al. first postulated the peripheral source of isoprene by measuring blood isoprene from vascular compartments of mechanically ventilated pigs[33]. Turner et al. found one isoprene deficient healthy adult[34]. While compared to isoprene normal adults, no correlations were seen between isoprene exhalation and fasting blood cholesterol profiles. King et al. also predicted its extrahepatic endogenous production via physiological modeling of exhaled isoprene dynamics[35]. The model reasonably explained the exercise driven immediate increase in breath isoprene. King et al. observed reduced (by a factor $\geqq 8$, while compared to healthy adults) levels of blood and breath isoprene in five late state muscle dystrophy patients and thereby, postulated its possible production in muscle[29]. Unterkofler et al. applied a two compartmental model to establish connection between endogenous production and metabolism of systemic VOCs and demonstrated that inhaled deuterated isoprene-D5 does not exhibit a peak at the beginning of exercise[36]. Although, we have reported pronounced effects of peripheral vasoconstrictions (in muscular compartments) during standing[37], while wearing medical face-masks[38] and also effects of natural menstrual cycle and daily oral contraceptive pills[39] on exhaled isoprene profiles, actual down-stream analysis to confirm the true origin in human and metabolic source of breath isoprene was completely missing. Recently, we executed breathomics, lipid profiling and gene expression analyses in an isoprene absent rare German adult and her isoprene deficient parents and sibling sister. Outcomes depicted no aberration in cholesterol levels and/or in gene expression of the mevalonate pathway enzymes and indicated a recessive inheritance of this healthy character[30]. Therefore, we questioned the putative human origin (hepatic cholesterogenesis) of exhaled isoprene that was proposed by Deneris et al. based on in vitro experiments in rat liver cells[40]. Nevertheless, a single rare case was insufficient for detailed downstream multi-omic analysis to determine the exact source. In a recent case study on one isoprene deficient American adult male and his blood-relatives by Harshman et al.[41] and in another study on an isoprene deficient Italian adult female and her blood-relatives by Biagini et al.[42] also demonstrated no relation of exhaled isoprene profiles to plasma cholesterol levels. They neither find any isoprene absent rare adult nor investigated the human cholesterol metabolism related gene expressions.

As a continuation of our precedent down-stream observations in the isoprene aberrated family[30], we conducted untargeted breathomics in clinical screening scenarios and discovered four more isoprene absent healthy German adults. Thus, to determine the exact metabolic origin, we have now performed multi-omic investigations involving, whole exome sequencing, breathomics and relevant serological analyses in these five rare adults (amongst 2000 recruited subjects) along with targeted sequencing of lead variants in blood-related isoprene deficient and unrelated isoprene normal healthy German adults. Our findings revealed only one shared homozygous (<1% prevalent in Europeans) IDI2 stop-gain mutation (at c.431 position) causing loss of enzyme active site and magnesium ion–cofactor binding sites in the isoprene absent adults. This prevented the conversion of isopentenyl diphosphate to dimethylallyl diphosphate (DMAPP) in cholesterol metabolism pathway. The IDI2 variant turned out to be heterozygous in isoprene deficient blood-relatives and absent in unrelated isoprene normal adults. Wild-type IDI1 and cholesterol metabolism related serological parameters were normal in all adults. Unlike other mammals, naturally IDI2 knocked-out pigs and bottlenose dolphins do not exhale isoprene. In humans, IDI1 is expressed highly in liver but the hepatocellular cytochrome-P450 enzymes immediately oxidizes isoprene. Skeletal muscles metabolize lipids (oxidize fatty acids, cholesterol) to produce energy, regulate intramyocellular signaling and integrity. Peroxisomes are metabolic organelle—mainly responsible for lipolysis. Peroxisomal beta-oxidation produces acetyl-CoA, i.e., channelled towards DMAPP production. Thus, human IDI2 determines isoprene production as DMAPP is the only source of isoprene and unlike plants, humans do not have isoprene synthase and its enzyme homologue. In humans, IDI2 is only expressed in skeletal-myocellular peroxisomes and instant spikes in isoprene exhalation during any muscle activity confirm the origin from muscular lipolytic cholesterol metabolism. We discovered the genetic origin of human isoprene production and related biochemical routes. We translated isoprene as the first breath VOC biomarker with well-defined endogenous origins and metabolic pathways. This knowledge will redefine clinical interpretations and applications of this non-invasive metabolic biomarker for various physio-metabolic and pathophysiological conditions.

## Results

**Distribution of exhaled alveolar and inhaled isoprene concentrations**. Out of 2000 recruitments from clinical screenings,

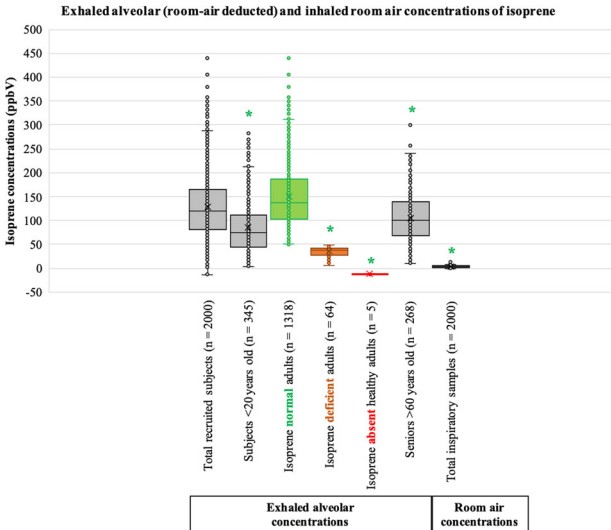

**Fig. 1 Distribution of exhaled alveolar and inhaled room air concentrations of isoprene.** Y-axis represents isoprene concentrations (ppbV). X-axis represents the total number of recruitments (n = 2000) along with different age groups/subgroups (with corresponding number of screened subjects) and total number of inspiratory samples (n = 2000). The adult group (aged 20–60 years) is sub-grouped into three colored boxes viz. isoprene normal (n = 1320; exhaled isoprene concentrations >50 ppbV), isoprene deficient (n = 64; concentrations <50 ppbV) and isoprene absent healthy (n = 5), respectively. Statistical significances of differences in isoprene concentrations between different age groups/subgroups were tested by means of one-way ANOVA on ranks (p-value ≤ 0.005). From all pairwise-multiple comparisons, statistically significant differences with respect to 'isoprene normal adults' are marked with green-colored '*'. The group containing the 'total recruited subjects' was not compared statistically to avoid overlaps of exhaled isoprene concentrations from different age groups and subgroups.

we obtained only five (3 females and 2 males) isoprene absent healthy adults (i.e., rare with <0.03% prevalence), 64 (39 females and 25 males) isoprene deficient (with low exhaled concentrations <50 ppbV) adults and 1318 (687 females and 631 males) isoprene normal (i.e., exhaled concentrations >50 ppbV) adults aged between 20 and 60 years. Besides, we obtained 345 (158 females and 187 males) subjects aged between 01 and 19 years and 268 (196 females and 72 males) subjects aged between 61 and 90 years. Correlations between biological age (year) and exhaled isoprene concentrations (ppbV) are presented in Supplementary Table 1.

Figure 1 represents the distribution of exhaled alveolar and inhaled room air concentrations (ppbV) of isoprene measured via real-time PTR-ToF-MS from 2000 recruited subjects. Exhaled isoprene concentrations are also presented within different age groups and subgroups. Statistically significant (p-value ≤ 0.005) differences between groups with respect to isoprene normal adults are indicated via green colored '*' and corresponding p-values are presented in Supplementary Table 2. Numerical source data on exhaled alveolar (room-air deducted) isoprene concentrations are presented in Supplementary Data 1.

Exhaled alveolar (corresponding room air deducted) isoprene concentrations (mean ± SD) in subjects aged <20 years, isoprene normal adults, isoprene deficient adults, isoprene absent adults and subjects aged >60 years were 86.51 ± 59.80 ppbV, 150.24 ± 65.13 ppbV, 35.27 ± 10.33 ppbV, −12.23 ± 0.91 ppbV (corresponds to 0.00 ppbV in breath, i.e., below LLOD and LLOQ) and 105.03 ± 49.70 ppbV, respectively. Room air concentrations were 3.80 ± 3.74 ppbV. The group of young

subjects (aged <20 years) had significantly (p-value < 0.001) lower isoprene levels than the group of seniors (aged >60 years).

Exhaled alveolar isoprene concentrations (corresponding room air subtracted) in rare adult-1's father (aged 60 years, German), mother (aged 60 years, German) and sibling sister (aged 30 years, German) were 15.86 ppbV, 17.54 ppbV and 27.24 ppbV, respectively. These three adults were isoprene deficient and healthy. Previous serological investigations showed normal lipid profiles in them[30].

**Demographic, serological, and breathomic parameters from rare adults.** Demographic data (age, gender, ethnic origin, health status and lifestyle habits), relevant serological parameters (viz. plasma lipid profile, bile metabolites and sex hormones) and exhaled concentrations of prime endogenous and exogenous VOCs from the five isoprene absent adults are presented in Table 1.

No considerable serological aberration was found in the rare adults. Exhaled endogenous VOCs (except isoprene) were also within the expected normal ranges. Exhaled exogenous VOCs were mainly related to lifestyle habits.

**Filtering strategies for the identification of candidate mutations.** Figure 2 represents the filtering strategies for the identification of candidate mutations following whole exome sequencing of isoprene-absent healthy German adults (n = 5) identifying heterozygous and homozygous variants across all known gene and protein coding regions. The filtering strategy for the detection of rare homozygous deleterious variants shared by all individuals are presented in 2a. Throughout the exome, over 63,000 variants passed the upstream bioinformatics pipeline to secure sufficient data quality and were present in at least one of the five isoprene-absent adults. As a recessive mode of inheritance was suspected, we next filtered for homozygous variants, resulting in roughly 25,000 remaining candidate variants. To justify biological significance, we only included mutations that resulted in changes on amino acid level, including frameshift, missense, nonsense or indel variants, lowering the number of potential candidates to around 9000. Knowing roughly the frequency of the investigated character (5 homozygotes out of 2000), we aimed to exclude variants with a reported population frequency of 15% or higher, being tantamount to roughly 1% homozygotes. Finally, out of those 556 rare homozygous variants, only one (IDI2, c.431 G > A) was shared by all five isoprene-absent adults. The detected mutation is listed in dbSNP (rs1044261) but no association towards isoprene metabolism was mentioned so far.

We further investigated if there are any heterozygous variants located within the mevalonate arm of cholesterol biosynthesis and steroid hormone metabolism pathway that might be related to the character (Fig. 2b). Again, out of the roughly 63,000 variants that passed upstream quality assessment, over 25,000 mutations resulted in frameshift, in-del, nonsense or missense mutations. We then analyzed the pathway genes ACAT2, HMGCS1, HMGCR, MVK, PMVK, MVD, IDI1, IDI2, FDPS1, GGPS1, FDFT1, SQLE, LSS and DHCR7, resulting in 12 variants that are present in at least one of the isoprene-absent adults. The above-listed genes are step-wise converting acetyl-CoA to cholesterol. However, no additional mutation was shared by all cases, except for an FDFT1 variant that is very common in the European population and can therefore be excluded. Corresponding amino acids, variants, impact, variant allele frequencies and prevalence (%) of homozygous and heterozygous variants in the European population (based on gnomAD) are presented in Table 1.

**Bidirectional targeted sequencing of the IDI2 mutational site.** We next aimed to investigate the inheritance pattern of the IDI2

**Table. 1 Demographic, serological, breathomic, and genomic parameters from rare adults.**

| Category | Parameter | Isoprene absent rare adults | | | | | Reference range | Method |
|---|---|---|---|---|---|---|---|---|
| | | 1 | 2 | 3 | 4 | 5 | | |
| **Demographic parameters** / Demography & lifestyle | ID | 1 | 2 | 3 | 4 | 5 | | |
| | Age (years) | 32 | 43 | 28 | 44 | 27 | | |
| | Gender | Female | Female | Female | Male | Male | | |
| | Ethnicity | German | German | German | German | German | | |
| | Clinical status | Healthy | Healthy | Healthy | Healthy | Healthy | | |
| | Special therapy/Diet | No | No | No | No | No | | |
| | Smoking/Drinking habit | No | Smoker | No | No | Alcohol (occation) | | |
| **Serological parameters** / Lipids | Cholesterol (mmol/L) | 4.4 | 4.6 | 3.9 | 4.0 | 4.9 | < 6.18 | CHOD-PAP/LiHeparin |
| | HDL (mmol/L) | 1.27 | 1.97 | 1.77 | 1.68 | 1.72 | 1.09 - 2.28 | PHOT/LiHeparin |
| | LDL (mmol/L) | 2.82 | 2.72 | 1.94 | 2.31 | 3.12 | 1.76 - 4.11 | PHOT/LiHeparin |
| | Lipoprotein a (g/L) | 0.5 | 2.15 | 0.04 | 0.05 | 0.07 | < 0.3 | NEPH/LiHeparin |
| | Triglyceride (mmol/L) | 0.67 | 0.63 | 0.75 | 1.46 | 0.84 | < 1.7 | GPO-PAP/LiHeparin |
| Metabolites | Total Bilirubin (µmol/L) | 9.49 | 7.21 | 12.5 | 13.3 | 38.2 | 1.71 - 20.5 | PHOT/LiHeparin |
| | Direct Bilirubin (µmol/L) | 1.35 | 0.93 | 3.11 | 2.45 | 4.74 | < 5.1 | PHOT/LiHeparin |
| | Indirect Bilirubin (µmol/L) | 8.14 | 6.28 | 9.39 | 10.9 | 33.5 | 3.4 - 12.0 | /LiHeparin |
| | 17β-Estradiol E2 (pmol/L) | 75.4 | 1101 | 504 | 129 | 18.4 | Female (premenopausal): 110 - 1468.4 Female (postmenopausal): 0 - 110 Male: 36.7 - 183.6 | ECLIA/Serum |
| Hormones | Progesteron (nmol/L) | 0.74 | 29.4 | 6.63 | 0.59 | 0.86 | Female (pre-ovulation): 3.18 Female (mid-cycle): 15.9 - 63.6 Male: < 3.18 | ECLIA/Serum |
| | Testosteron (ng/mL) | 0.22 | 0.37 | 0.44 | 6.03 | 4.44 | Male: 2.5 - 9.5 Female: 0.1 - 0.9 | ECLIA/Serum |
| **Breathomic parameters** / Endogenous VOCs | Acetone (ppbV) | 395.77 | 588.61 | 409.76 | 1461.06 | 821.53 | 250 - 1000 | PTR-ToF-MS |
| | **Isoprene (ppbV)** | **0.00** | **0.00** | **0.00** | **0.00** | **0.00** | **100-300** | |
| | Acetic acid (ppbV) | 9.44 | 18.13 | 14.97 | 18.88 | 16.76 | 5-50 | |
| | Butyric acid (ppbV) | 0.96 | 1.54 | 0.90 | 1.86 | 1.40 | 0.1 - 5 | |
| | Dimethyl sulfide (ppbV) | 12.95 | 9.84 | 39.49 | 178.20 | 2.00 | 250 - 1000 | |
| | Acetaldehyde (ppbV) | 38.82 | 33.79 | 58.77 | 126.97 | 63.38 | 10 - 180 | |
| | Crotonaldehyde (ppbV) | 0.23 | 0.57 | 0.38 | 0.51 | 0.32 | 0.1 - 1 | |
| | Pentanal (ppbV) | 22.27 | 58.32 | 57.80 | 60.10 | 17.89 | 5 - 100 | |
| Exogenous VOCs | Acetonitrile (ppbV) | 4.30 | 123.21 | 5.70 | 8.40 | 3.98 | Smoking related | |
| | Ethanol (ppbV) | 6.93 | 82.28 | 49.30 | 10.04 | 493.40 | Drinking related | |
| | Benzene (ppbV) | 0.19 | 2.40 | 0.92 | 1.58 | 0.78 | Enviornmental | |
| | Toluene (ppbV) | 1.32 | 1.81 | 1.45 | 2.07 | 1.71 | Expouser | |
| | Limonene (ppbV) | 133.97 | 17.53 | 35.91 | 30.17 | 12.42 | Diatery | |

**Genomic parameters** — Shared homozygous mutation with <1% population prevalence

| | Genes | Amino acids | Variants | Impact | Variant allele frequency | | | | | Reference | Heterozygous % | Homozygous % |
|---|---|---|---|---|---|---|---|---|---|---|---|---|
| | | | | | 1 | 2 | 3 | 4 | 5 | European population (gnomAD) | | |
| | ACAT2 | p.K211R | c.632 A > G | Missense | 0.00 | 100.00 | 41.82 | 100.00 | 43.08 | | 57.66 | 14.47 |
| | HMGCR | p.T582M | c.1745C>T | Missense | 96.23 | 0.00 | 0.00 | 0.00 | 0.00 | | 16.41 | 0.65 |
| | HMGCR | p.G580fs* | c.1737_1752del CGGACAGACGGATGGA | Frameshift deletion | 37.08 | 45.00 | 40.98 | 0.00 | 0.00 | | 25.21 | 2.72 |
| | MVK | p.S52N | c.155 G > A | Missense | 0.00 | 0.00 | 0.00 | 100.00 | 52.00 | | 13.11 | 0.82 |
| | PMVK | p.N14S | c.41 A > G | Missense | 0.00 | 0.00 | 39.66 | 45.28 | 0.00 | | 6.43 | 0.17 |

Genomic method / Reference: Whole human exome sequencing (All Exon V8, end seq. 2×150 bp and 50x average on target)

**Table 1 (continued)**

| | | | | Isoprene absent rare adults | | | | | |
|---|---|---|---|---|---|---|---|---|---|
| *MVD* | Splicing | c.70+5 G > A | Splicing | 100.00 | 0.00 | 0.00 | 50.67 | 0.41 | 0.00 |
| ***IDI2*** | **p.W144\*** | **c.431 G > A** | Stop gain | **100.00** | **100.00** | **100.00** | **100.00** | **6.95** | **0.23** |
| *FDFT1* | p.H76_S77del | c.226_231delCACTCC | Deletion | 100.00 | 100.00 | 100.00 | 100.00 | 93.86 | 38.69 |
| *LSS* | p.R95Q | c.284 G > A | Missense | 44.34 | 0.00 | 46.67 | 0.00 | 15.77 | 1.27 |
| *LSS* | p.H230R | c.689 A > G | Missense | 56.52 | 0.00 | 49.40 | 0.00 | 15.86 | 1.28 |
| *LSS* | p.L562V | c.1684T>G | Missense | 100.00 | 100.00 | 100.00 | 55.70 | 65.39 | 18.62 |
| *DHCR7* | p.M389T | c.1166 T > C | Missense | 0.00 | 100.00 | 100.00 | 100.00 | 95.36 | 39.43 |

Demography (age, gender, ethnic origin, and health status), lifestyle (smoking, drinking habits and therapy and special diet), relevant serological parameters (plasma lipid profile, bile metabolites, and sex hormones) and exhaled concentrations of prime endogenous and exogenous VOCs from the five rare adults are presented. Corresponding reference range and applied methods are presented. Aberrated parameters are marked in bold. Gene names are written in italics.

c.431 G > A variant. Therefore, confirmatory targeted bidirectional Sanger sequencing was performed for all isoprene absent adults (Fig. 3). As expected, the results demonstrated a homozygous mutation in the respective position in all isoprene-absent specimens, confirming the previous whole exome sequencing findings. Family members of isoprene absent adult-1, who demonstrated <30 ppbV of isoprene in exhaled breath, were also checked, revealing a heterozygous *IDI2* genotype in her mother, father and sibling sister. Two unrelated healthy adults with normal isoprene profiles were sequenced as controls. Those control participants did not show the *IDI2* c.431 G > A variant, neither homozygous nor heterozygous.

**IDI2 gene and protein structure.** We finally interrogated the UCSC Genome Browser and UniProt databases to elucidate the biological significance of the detected *IDI2* c.431 G > A variant. The *IDI2* gene is located on the short arm of chromosome 10 and consists of five exons, with coding sequences in exons two to five (Fig. 4a). The mutation of interest is located at the beginning of exon five and results in a stop gain at p.W144*, accounting for a truncated mRNA (Fig. 4a). This genomic area is highly conserved, indicating a crucial and ubiquitous role of the transcriptional site.

The wild-type IDI2 protein spans 227 amino acids (aa), with a large hydrolase domain spanning aa 49 to 199 (Fig. 4c). There are two active sites of the enzyme, one at the N-terminus of the hydrolase domain and the other one at aa 148, only four aa downstream from the detected p.W144* variant (Fig. 4d). Due to the truncated mRNA transcript, the second enzyme active site is deleted in the five individuals sharing the c.431 G > A mutation, likely resulting in impaired or absent protein function. In addition, the loss of the C-terminal part of *IDI2* exon 5 also results in the deletion of two out of four magnesium binding sites, which is a required cofactor for the enzyme's function.

**Discussion**

For the last 39 years, many have erroneously regarded hepatic cholesterogenesis (producing >90% of human cholesterol) as the prime origin of human exhaled isoprene. This believe was based on in vitro synthesis of isoprene from DL-mevalonate by utilizing a rat liver cytosolic fraction[40]. Consequently, various physio-metabolic and clinical conditions driven interesting differences (cross-sectional) and/or changes (longitudinal) in isoprene exhalation could not be explained via the well-known/established effects of those conditions on hepatic cholesterogenesis. As a result, breath isoprene could not step into routine clinical practice as a noninvasive biomarker. However, articles by Miekisch et al.[33], Turner et al.[34], and King et al.[35] indicated extrahepatic production of this VOC and physiological modeling approaches under exercise hinted toward a probable muscular origin[29,36]. Despite a few studies reported the rare presence of isoprene absent adults[25,31,34,40–43], until 2021, there was no down-stream evidence available to completely disregard the convention. In 2021, we finally disqualified the putative origin of breath isoprene[30] and here, we have discovered the actual origin of human exhaled $C_5H_8$ by multi-omic analysis of genes and metabolites.

Distribution of exhaled isoprene concentrations from 2000 screened subjects reconfirmed previously reported age dependency[23–25,30] of its exhalation. Complete absence of exhaled isoprene in the rare adults is caused by the shared homozygous *IDI2* variant (stop-gain mutation at c.431 position). This mutation was the only variant in whole exome sequencing that was present in all specimens and met inclusion criteria. Functional aberrations of the enzyme active site and metal–cofactor binding

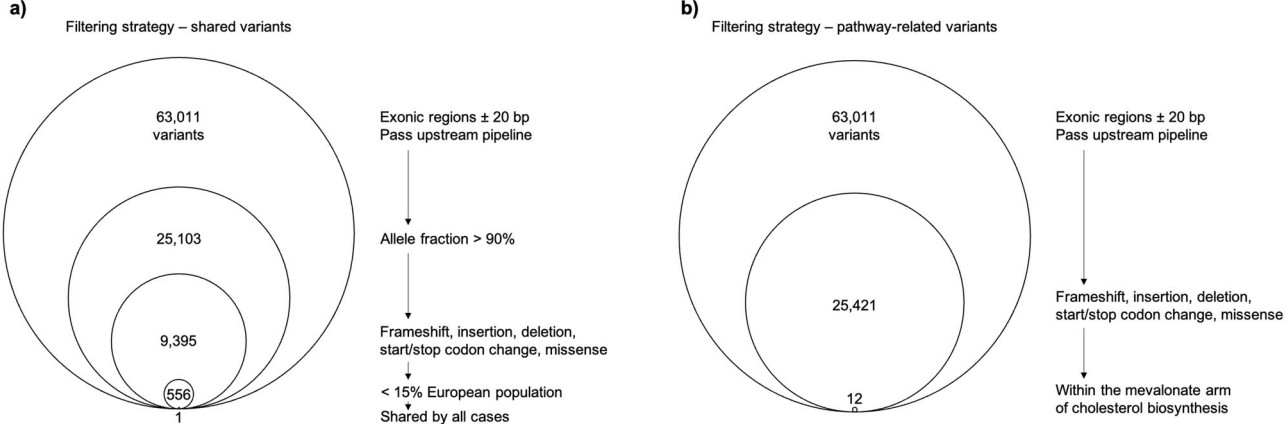

**Fig. 2 Filtering strategies for the identification of candidate mutations following exome sequencing of isoprene absent healthy adults. a** Filtering strategy for the detection of rare homozygous deleterious variants shared by all individuals. **b** Filtering strategy for the characterization of variants located within the mevalonate arm of cholesterol biosynthesis and steroid hormone metabolism. including *ACAT2, HMGCS1, HMGCR, MVK, PMVK, MVD, IDI1, IDI2, FDPS1, GGPS1, FDFT1, SQLE, LSS,* and *DHCR7.*

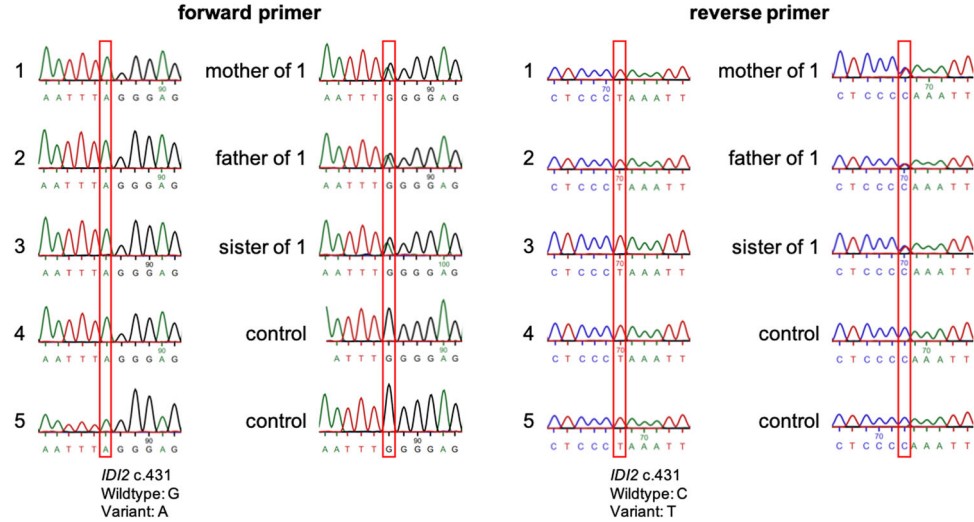

**Fig. 3 Bidirectional targeted sequencing of the *IDI2* mutational site.** Sanger sequencing chromatograms using forward (left) and reverse (right) primers to analyze the *IDI2* c.431 position. DNA nucleotide bases are assigned as A, T, G and C and are presented with green, red, black and blue colored peaks. respectively. Isoprene absent rare adults (1–5) demonstrate homozygosity while blood-related family members (isoprene deficient) of rare adult 1 were confirmed heterozygous. The mutation is absent in unrelated adult controls. Red boxes indicate the variant genomic site c.431.

sites likely arise through this mutation. Looking at the isoprene exhalation in our previous study[30], we assumed that the inheritance of the character (isoprene absence) has a recessive trait. Here, the heterozygous presence of the *IDI2* c.431 G > A variant in the isoprene deficient healthy parents and sibling sister of rare adult-1 and absence of this *IDI2* mutation in unrelated healthy adults (isoprene normal) genetically confirmed our previous assumptions. It is possible, that a certain number of subjects from our clinical studies were first- or second-degree relatives. As per ethical obligations, we were not allowed to use/disclose such non-anonymous/identifiable information and/or to look for relationships between participants during their recruitments in clinical screening studies. Therefore, we did not consider such information during data analysis in our present study. Within this study, we were allowed to contact (only via formal letters of invitation) the adult family members of the isoprene absent adults. At the best of our knowledge, the five isoprene absent adults were unrelated to each other. Except the presented family members of the isoprene absent adult-1, no other blood relatives of the remaining rare adults were involved within this study.

In this study, the prevalence (<0.25%) of isoprene absent adults closely mirrored the actual homozygous prevalence (0.23%) of the *IDI2* rs1044261 variant (p.Trp144Stop) in the EU population. A broader approach of parallel *IDI2* sequencing in combination with breath analysis, however, is necessary to definitely confirm the relation between the amount of exhaled isoprene and *IDI2* genotype. Nevertheless, such approach is considerably resource demanding in a large scale and remained beyond our immediate aim and present resource capacity. The overall expression (homozygous and heterozygous) of the mutated *IDI2* is different in other ethnic origins (Supplementary Table 3) and therefore, the observed age distributions of isoprene, deficiency and/or absence may differ amongst another ethnicity/population. Besides, the cut-off limit of isoprene deficiency in adults was set by us to < 50 ppbV to define a clear cut-off beyond the inter-individual and physiological variations (normal range of exhaled isoprene 80–350 ppbV). Exhaled isoprene concentrations may vary by 5–25 ppbV in an individual simply due to his/her normal physiological variations in respiratory and hemodynamic parameters, natural menstrual rhythms and/or oral contraception and

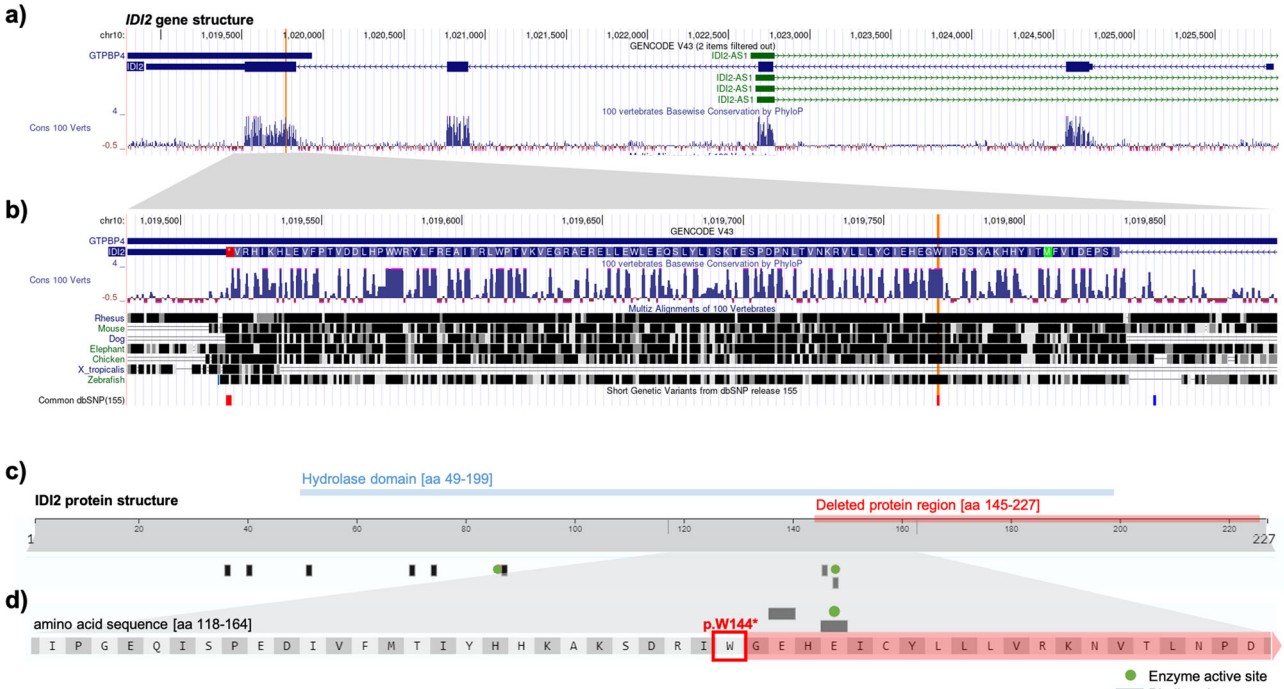

**Fig. 4 *IDI2* gene and protein structure. a, b** The gene structure was retrieved from the UCSC Genome Browser website (http://genome.ucsc.edu) using the GRCh38/hg38 assembly. **a** The full gene structure of *IDI2* with the site of the c.431 G > A point mutation marked in orange. **b** Enlargement of the *IDI2* exon 5 coding regions. **c, d** The *IDI2* protein structure was retrieved from the UniProt database (http://uniprot.org) and the area of the hydrolase domain was added in blue. The deleted protein region following the p.W144* mutation is marked in red. Enzyme active sites and binding sites are depicted as black boxes and green circles. respectively. **c** Full protein structure. **d** Enlargement of amino acids 118–164. including the mutation site marked with a red box.

menopause[23,37,39,44] etc. Therefore, the cut-off <50 ppbV allowed us to determine the clearly isoprene deficient adults—beyond his/her physio-metabolic fluctuations.

While the expression of human *IDI1* is conserved in various tissues and high within the mitochondria and proteasome of the hepatocytes, its divergent isoform *IDI2* is highly expressed only within the peroxisome of the skeletal myocytes[45,46]. Human *IDI1* is poorly expressed in skeletal muscle. *IDI1* and/or *IDI2* catalyze the isomerization of isopentenyl diphosphate [($C_{14}$)IPP] to the highly nucleophilic dimethylallyl diphosphate [($C_{14}$)DMAPP]. In humans, the conversion of IPP to DMAPP takes place in two metabolic paths— during cholesterol biosynthesis in the endoplasmic reticulum of hepatocytes and during lipid catabolism (involving cholesterol metabolism) in the peroxisome of skeletal myocytes[47,48]. Only DMAPP (not IPP) is converted to $C_5H_8$ via *isoprene synthase* (IspS) enzyme in plants[49–51]. As humans do not have *isoprene synthase* and bioinformatic sequence alignment (whole exome/functional domain based) via BLAST search tool[52] did not locate human enzyme homologs of *isoprene synthase*, the wild-type *IDI1* and *IDI2* genes (and related proteins) should serve as the determinant for human isoprene production from those two aforementioned metabolic paths.

Besides humans, while looking at other terrestrial and marine mammals, we observed interesting facts upon breath isoprene. In a recent pre-clinical study, mass-spectrometry based untargeted profiling of exhaled VOCs in spontaneously breathing awake healthy and/or influenza A virus infected pigs, we could not detect breath isoprene[53]. On the other hand, in pre-clinical breathomic studies on goats and on cattle, we observed significant concentrations of breath isoprene from both ruminants[54,55]. Via mass-spectrometry based comprehensive screening of exhaled metabolites from bottlenose dolphins, Aksenoy et al. did not detect any trace of isoprene[56]. Our present search in the Ensembl genome database[57] and EMBL-EBI resource[58] showed that *IDI2*

is not at all expressed in pigs and in bottlenose dolphins but is well expressed in goats and cattle, underlining functional *IDI2* as discriminator between isoprene presence and absence. *IDI1* is ubiquitously expressed in many tissues in all these animals.

Previously, via microextraction-coupled mass-spectrometric measurements of headspace of arterial and venous blood samples collected from mechanically ventilated humans and pigs, we observed extremely low (up to 10-fold lower than in human) isoprene concentrations within the portal and mixed venous blood of pigs[33]. Such tiny fraction may be washed-out (i.e., stored previously) and/or produced via minimal *IDI1* activity in the peripheral compartments. Nevertheless, as soon as the blood crossed the hepatic circulation, isoprene concentrations were diminished in hepatic venous samples – most likely due to a high rate of isoprene metabolism in pig liver. Similar to pigs and dolphins, only the *IDI1* is expressed in rats and mainly within the liver. Most likely, due to a low (compared to higher mammals) isoprene oxidation rate in rat liver[59], Deneris et al. had detected a certain fraction of isoprene in rat liver cytosol in vitro. They suggested that isoprene could be produced in rat liver via non-enzymatic degradation of IPP and/or DMAPP and postulated in general that breath isoprene is linked to hepatic cholesterogenesis[40]. While the pre-clinical finding of Deneris et al. was correct, the general inference drawn on the origin of human breath isoprene based on those outcomes from rats was wrong. In human hepatocellular microsomes, the complex cytochrome P450 enzyme system immediately oxidizes isoprene and isoprene monoepoxides to avoid hemiterpene toxicity[60]. The oxidation rate in human liver microsomes is magnitudes higher compared to rats[59]. Due to such high isoprene oxidation rate, hepatic cholesterogenesis is insufficient to contribute any considerable concentration of isoprene to human exhalation.

In the present study, despite normal plasma lipid profiles, bile substrates, sex-hormones, and wild-type *IDI1* in rare adults and

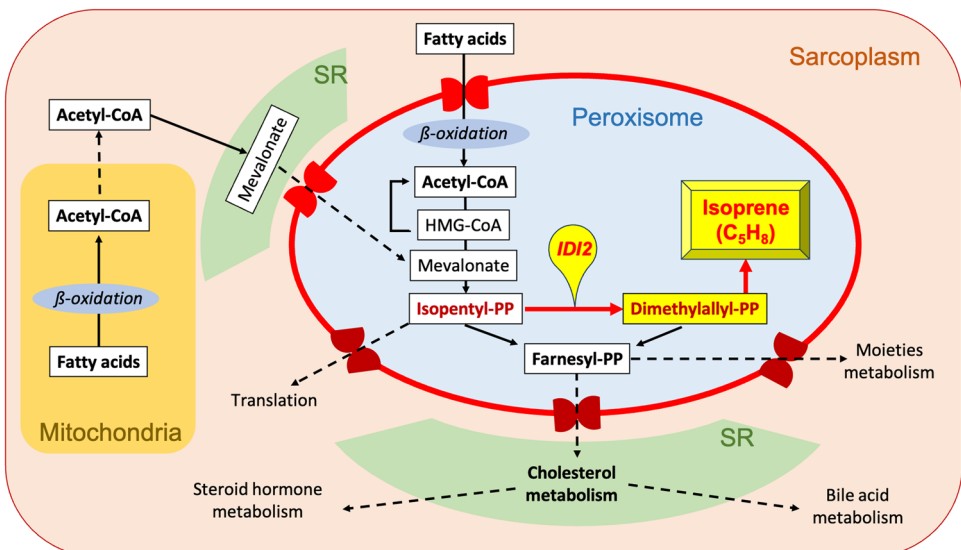

**Fig. 5 Isoprene production pathway with *IDI2* activity in human skeletal-myocyte.** The pathway is based on the results of this study. The pathway depicts the potential contributions of myocellular organelles, e.g., sarcoplasm, sarcoplasmic reticulum (SR), mitochondria and most importantly the peroxisomes. Human *IDI2* is only and highly expressed within the myocellular peroxisomes. Peroxisomes are metabolic organelle—mainly responsible for lipolysis. Here, acetyl-CoA is produced via beta-oxidation of fatty acids within peroxisomes. Acetyl-CoA is then channeled towards farnesyl-PP (FPP) production. One step before the FPP generation, isopentyl-PP (IPP) is converted into its active isoform dimethylallyl-PP (DMAPP) by the *IDI2* activity. *IDI2* does not convert DMAPP back to IPP. In human, DMAPP is the only source of isoprene. Unlike plants, humans lack *isoprene synthase* and its enzyme homologue. Thus, *IDI2* determines human isoprene production in skeletal-myocytes. Farnesyl-PP exits peroxisomes and enters other organelles to execute in other metabolic pathways, e.g., cholesterol metabolism in the SR.

blood-related family members, significant aberrations in their isoprene exhalations confirmed our previously suggested[30] independence of human exhaled isoprene from hepatic cholesterogenesis and related principal pathways. Therefore, due to the absence of functional *IDI2* expressions in these isoprene aberrated adults, the source of human breath isoprene should mainly be directly attributed to muscular metabolic activity and not to hepatic cholesterol biosynthesis. The isoprene absent adults constantly inhaled low concentrations of ambient isoprene from the inspired air but they never exhaled any tracible fraction of this VOC. This indicates an uptake/partial storage and/or metabolism of ambient isoprene in these individuals.

Skeletal muscles represent around 40% of adult human body mass and predominantly utilize glucose and lipids to produce energy, regulate intramyocellular signaling and integrity[61,62]. Insulin governs the balance between glucose and fatty acid metabolism in muscle[63] and peroxisomal beta-oxidation senses intracellular fatty acids and regulates lipolysis[64]. Besides mitochondrial oxidation, peroxisomal beta-oxidation of very long-chain fatty acids, long-chain fatty acids and dicarboxylic acids produces acetyl-CoA. Acetyl-CoA is channeled towards farnesyl diphosphate (farnesyl-PP) production inside the peroxisomes[48]. All enzymes (except 3-hydroxy-3-methylglutaryl-CoA reductase/ HMGCR) step-wise converting acetyl-CoA to farnesyl-PP contain functional peroxisomal targeting signals and at the second last step of this pathway, IPP is converted to DMAPP via the IDI2 enzyme as only *IDI2* is highly expressed here. Farnesyl-PP exits the peroxisomes to execute various metabolic processes in other cellular organelles (Fig. 5).

Any kind of muscle movement/activity immediately gives rise to breath isoprene[13,65]. Due to its low aqueous solubility and high volatility, isoprene is positively related to cardiac output and negatively related to minute ventilation[66,67]. Both low-intensity and exhaustive exercise demonstrated an instant and profound increase in exhaled alveolar isoprene concentrations at the initial warm-up phase (that increases muscle perfusion) followed by

gradual decrease with increasing work-load, which indicates its possible production and washout from the active muscle compartments[14,65,68]. Exercise immediately increases skeletal muscular lipolysis, fatty acid transport from plasma to sarcoplasm and triglyceride hydrolysis to compensate energy demand. Thus, our present findings ascertain that isoprene is potentially originating from lipolysis in the skeletal muscle and wild-type *IDI2* denominates the presence of isoprene in exhaled human breath and also acts as the rate limiting factor for endogenous isoprene production.

Although we were able to detect the *IDI2* mutation in PBMC, *IDI2* gene and protein expression are skeletal muscle specific. Previously we observed *IDI1* but not *IDI2* gene expression in PBMC of healthy adults with absence, deficiency, and normal breath isoprene[30]. To assess the biological consequences of the *IDI2* variant, gene, and protein expression studies would require the collection of muscle biopsies of the affected adults. Those investigations are, however, behind the scope of the present study and limited by ethical considerations. In vitro experiments using muscle tissue or cell lines might shed light on the metabolic and functional background of isoprene synthesis, metabolism, and downstream function. Investigation of relationship between *IDI2* gene and protein expressions and isoprene exhalation under exercise was beyond the scope of the present study design but may reveal interesting insights in future studies.

We discovered the genetic origin of human breath isoprene production and related biochemical routes (Fig. 5). The rare character of isoprene absence/deficiency in healthy human adults is autosomal (10th chromosome, locus: 10p15.3) recessive. We translated isoprene as the first breath VOC biomarker with well-defined down-stream endogenous origin and metabolic pathways. This knowledge will redefine the clinical interpretations of this noninvasive biomarker for various physio-metabolic, pathophysiological, and inherited conditions. We assume that the presence of DMAPP may not be essential for healthy human life as all principal pathways converting acetyl-CoA to farnesyl-PP are

associated with lipid and cholesterol metabolism can utilize IPP to execute them normally. Already reported endocrine regulation[39] and age dependency[23,24,30] of isoprene exhalation indicates new research scopes of *IDI2* activity in human aging (Supplementary Table 1), muscle mass development and related conditions. Similarly, further investigation of *IDI2* gene and protein expressions in skeletal muscle tissue along with breath isoprene expressions under various exercise trainings and in individuals with muscle dystrophy and risk of rhabdomyolysis, e.g., under statin interventions or injury may reveal unexplored frontiers in sports/fitness and musculoskeletal medicine and inter-organ metabolic cross-talk.

In order to understand the evolutionary significance of human *IDI2* gene and the presence/rationales of isoprene production pathway in human, we need further multi-omic based system-wide evaluation from the genome to up-stream cascades. Tissue specific gene and protein expression followed by transcriptomics and proteomics may lead us to the actual enzymatic and meta-bolic significances (DMAPP to isoprene) that are linked to the last *IDI2* exaptation taken place ~70 million years ago for an unknown function. Inducible IDI2 deficiency in animal models could also shed light on the kinetics and physiological back-ground of isoprene metabolism. Further investigations should screen a large number of isoprene deficient subjects for *IDI2* allele frequencies to realize its exact genetic correlation(/predisposition) factor with breath isoprene expressions. In order to apply iso-prene or any other endogenous biomarker to routine clinical practice, well-defined down-stream origins and pathways should be addressed.

## Methods

**Study design.** Based on our previously observed rare character of exhaled isoprene aberrations (absence and deficiency) in healthy adults[30], we aimed to further investigate the actual reason(s) at the very down-stream level and thereby, to discover the principal human origin of this endogenous hemiterpene.

In adults, exhaled alveolar isoprene concentrations of 100–300 ppbV is regarded as the normal range and ±50 ppbV is regarded as the limit of normal physio-metabolic fluctuations. Exhaled alveolar concentration of 00 ppbV is regarded as isoprene absence and <50 ppbV is considered as significant isoprene deficiency. In adults, exhaled isoprene concentrations between.

In order to find adults with isoprene aberrations, we re-evaluated the isoprene exhalations in 2000 human subjects (aged between <1 and 100 years) from 15 consecutive clinical breath screening studies by applying real-time mass-spectrometry (proton transfer reaction—time of flight—mass spectrometry/ PTR-ToF-MS). All studies were conducted at the University Medicine Rostock, 18057 Rostock, Germany in accordance with the amended Declaration of Helsinki guidelines. Ethical approvals from the Institutional Ethics Committee (IEC, University Medicine Rostock, Rostock, Germany) and signed informed consents from all participants were obtained prior to participa-tion. In case of infants, both parents provided signed consents. Within these studies, healthy volunteers were recruited during physiological, metabolic, exercise and dietary/nutritional mon-itoring (Ethical approval numbers: A2011-67, A2012-0103, A2014-0037, A2015-0008, A2015-0076, A2018-0025, and A2020-0300) and both healthy and sick subjects were recruited during pathophysiological and/or therapeutic monitoring (Ethical approval numbers: A2012-0103, A2012-0071, A2015-0043, A2017-0106, A2018-0097, A2019-0040, A2020-0085, and A2021-0012). In the present study, ethical approval no. A2021-0012 is assigned to the investigations involving venous blood collection, peripheral blood mononuclear cells isolation, DNA

isolation, multi-omics (breathomics, untargeted and targeted genomics) and serological metabolites analysis in healthy isoprene aberrated and isoprene normal adults.

After finding the isoprene absent rare adults, we executed multi-omic investigations of shared down-stream aberrations in genes and relevant metabolites among such adults, in blood-relatives and in unrelated healthy controls. Therefore, at first, we conducted whole exome sequencing to identify unknown homozygous variants shared by the rare adults. Then the shared mutations were checked via bidirectional Sanger sequencing in blood-related (isoprene deficient) and unrelated (isoprene normal) healthy adults. Plasma lipids, metabolites and hormones related to cholesterol metabolism were also checked serologically in all these subjects.

Due to ethical reasons, we were allowed to conduct genomics only in adults. Amongst the isoprene absent rare adults, we could only access the blood-related adult family members (parents and sibling sister) of the rare adult-1.

**Breath sampling, VOC data analysis, and quantification.** All spontaneously breathing subjects maintained a defined posture[37] and performed oral breathing[44] via customized mouthpiece[69] or mask by following our state-of-the-are sampling protocol[67]. Continuous side-stream sampling (flow: 20–100 ml/min) from the mouthpiece or mask were performed via the heated (75–100 °C) transfer-line of a PTR-ToF-MS-8000 or a PTR-ToF-MS-1000 (Ionicon Analytik GmbH, Innsbruck, Austria) under pre-optimized experimental conditions[70,71]. Most importantly, PTR time-resolution of 200 ms, drift-tube temperature of 75 °C, voltage of 610 V and pressure of 2.3 mbar were used to reach the optimal E/N ratio of 139 Td[23,30,72]. After automatic recording of a data file/min the mass scale was recalibrated based on masses namely, 21.0226 ($H_3O^+$-isotope), 29.998 ($NO^+$) and 59.049 (protonated $C_3H_6O$).

We used a PTR-MS viewer software (version 3.228) to process raw data. VOC data were measured continuously in counts per second (cps). Measured VOC counts were normalized to primary ion ($H_3O^+$) counts. Breath-resolved assignment of expiratory (alveolar/end-tidal) and inspiratory (room air) phases were executed via custom-made 'breath tracker' algorithm[73,74]. Here, we used an endogenous VOC (e.g., acetone) with orders of magnitudes higher concentration in exhalation than in room air as the tracker mass.

Measured VOCs were quantified either via reaction rate coefficients ($k$-rates) between VOC and primary ion (at the E/N ratio of 140 Th) or via multi-component VOC standard mixture under matrix adapted conditions (breath humidity) by using a liquid calibration unit (LCU, Ionicon Analytik GmbH, Innsbruck, Austria)[72,75]. Isoprene was quantified via LCU based calibrations.

**Venous blood sampling.** A total of 50 ml of antecubital venous blood was collected from each subject by a skilled and licensed physician for serological and genetic analysis.

**Plasma metabolite analysis.** Plasma lipids (total cholesterol, lipoproteins a, high density lipoproteins (HDL), low density lipo-proteins (LDL) and triglycerides), bile substrates (total-, direct- and indirect bilirubin) and sex-hormones (estrogen, progesterone and testosterone) were analyzed at the central laboratory (Institute of Clinical Chemistry and Laboratory Medicine, University Medicine Rostock) via conventional methods (listed in Table 1).

**Isolation of peripheral blood mononuclear cells.** A total of 45 ml of venous blood was collected from all participants and mixed 1:1 with pre-warmed PBS (PAN-biotech, Aidenbach,

Germany). Density gradient centrifugation ($1200 \times g$, 12 min, 4 °C, brake 0) using PAN-coll separation solution (PAN-biotech) was carried out to separate peripheral blood mononuclear cells (PBMC). Isolated PBMC populations were washed twice ($180 \times g$, 10 min) in PBS and cell numbers were determined. Cell pellets were stored at −80 °C for subsequent analyses.

**Exome sequencing**. Cell pellets were thawed and genomic DNA was isolated using the NucleoSpin® Tissue kit (Macherey-Nagel, Düren, Germany) according to the manufacturer's instructions. Samples were sequenced via INVIEW Human Exome sequencing service (Eurofins Genomics, Ebersberg, Germany). The QIAGEN Clinical Insight Interpret platform (Qiagen, Hilden, Germany) was used to identify rare and unknown variants, using the following filtering criteria: only variants in exonic regions or variants that are ±20 bp flanking those regions; variants pass upstream pipeline filtering; variants with call quality ≥20; variants with allele fraction ≥90; variants that result in frameshift or in-frame indel or start/stop codon change or missense or nullizygous; variants with homozygous population frequency <1% in Europeans or no population data available.

**Bidirectional Sanger sequencing**. The PCR was carried out in a final volume of 25 μl containing 100 ng of genomic DNA, 0.4 μM of each primer (Eurofins Genomics), 250 μM of each dNTP (Agilent Technologies, Santa Clara, USA), 5 μl 5x Hi-Fi Reaction buffer (Meridian Bioscience, Cincinnati, OH, USA) and 0.5 μl VELOCITY™ DNA polymerase (Meridian Bioscience) using the following primers: F, AATTCTGTGTTTTACATTAGCGTTG; R, CTGGGACAGGTAGAGGATGCT. The PCR conditions were 2 min at 98 °C followed by 35 cycles of 30 s at 98 °C, 30 s at 57 °C and 15 s at 72 °C with a final step of 5 min at 72 °C. The products were further processed via QIAquick PCR Purification Kit (Qiagen) according to the manufacturer's instructions. Purified PCR products were sequenced using the Mix2Seq sequencing service (Eurofins Genomics).

**Statistics and reproducibility**. Exhaled alveolar and inspired room air isoprene concentrations from a minute of steady spontaneous breathing (with normal respiratory rate of 10–14 breaths/min) were considered for quantification. Measurement was repeated three times in each subject. Due to non-parametric distributions, median values were used for statistical analysis.

Statistical significances of differences in isoprene concentrations between different age groups/subgroups were tested by means of Kruskal-Wallis one-way ANOVA on ranks (Kolmogorov-Smirnov test for normality followed by the multiple comparisons via post-hoc Dunn's method at $p$-value ≤ 0.005) in SigmaPlot version 14. Here, groups with young subjects (aged <20 years, $n = 345$), isoprene normal adults (aged 20–60 years, $n = 1318$), isoprene deficient adults (aged 20–60 years, $n = 64$), isoprene absent adults (aged 20–60 years, $n = 5$), seniors (aged >60 years, $n = 268$) and total inspiratory samples ($n = 2000$) were compared. From all pairwise-multiple comparisons, statistically significant differences with respect to 'isoprene normal adults' were of study importance (Fig. 1). Detailed data on each group size (N), group median, difference in ranks and corresponding $p$-values are presented in Supplementary Table 1. In order to avoid overlaps of exhaled isoprene concentrations from different age groups and/or subgroups, the 'total recruited subjects' was not compared statistically (Fig. 1). Comparison of differences (with corresponding p-value) between the young subjects (aged <20 years) and seniors (aged >60 years) are also presented in the results.

Overall and group-wise correlations between subject's biological age (year) and exhaled isoprene concentrations (ppbV) were

assed via Spearman correlation test (due to non-parametric distribution of data).

**Reporting summary**. Further information on research design is available in the Nature Portfolio Reporting Summary linked to this article.

## Data availability
All disclosable data are available in the main text or the supplementary materials. Raw and processed experimental data (i.e., not under ethical restrictions) is available from the corresponding author upon reasonable request. However, exome sequencing raw data cannot be made publicly available due to the General Data Protection Regulation (GDPR) of the European Union. Due to the small number of rare individuals, this potentially identifying and sensitive patient information could compromise the privacy of research participants.

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

## Acknowledgements

We thank all participants for their voluntary participation in this study. We thank Dr. Fuchs (ROMBAT Group) for supporting us with the timely arrangements of consumables and materials ordering process. We thank Dr. Brock for collection of blood samples and for supporting the PBMC analysis. We thank our colleagues (Clinical chemists: Dr. Holdt, Dr. Bastian and Dr. Schubert) from the central laboratory (Institute of Clinical Chemistry and Laboratory Medicine) for timely serological analysis of all blood samples. The study was partly supported by the following research grants: European Union fund for regional development EFRE (JKS, WM). EU Horizon-2020 grant H2020-PCH-HEARTEN project 643694 (W.M., J.K.S., P.S.). European Union's Horizon 2020 Marie Skłodowska-Curie research and innovation programme 674911-IMPACT (J.K.S., W.M., P.S.). University Medicine Rostock's FORUN programme (2018) 889003 (P.S.).

## Author contributions

P.S. conceived the idea and with J.K.S. and W.M. conceptualized the study. P.S., A.R., W.M., and J.K.S. designed the study. P.S., A.R., C.J., W.M., and J.K.S. developed the analytical methods. P.S. and A.R. recruited volunteers and performed experiments. P.S. and A.R. analyzed data. prepared and visualized results. P.S., A.R., C.J., W.M., and J.K.S. interpreted outcomes. W.M. and J.K.S. provided all resources. W.M., C.J., and J.K.S. supervised the entire project. P.S and A.R. wrote the original draft. W.M., C.J., and J.K.S. reviewed and edited the original draft. All authors approved the final version of the manuscript.

## Funding

## Competing interests

The authors declare no competing interests.
