## [Peer Review File · Communications Biology]

Reviewers' comments:

Reviewer #1 (Remarks to the Author):

see attachment

Reviewer #2 (Remarks to the Author):

The manuscript submitted by Sukul et al. has examined the metabolic origin of endogenous isoprene in humans by multi-omic analysis. The submitted work demonstrates that IDI2 plays a crucial role in the determination of isoprene production in humans. In general, the study is carefully designed and presented clear and significant results. The present study has made an important step toward understanding the mechanisms underlying human isoprene production. Thus, I certainly feel that the paper is important to the field. Meanwhile, there are several relatively minor concerns with the manuscript as written that must be addressed to help precise comprehension of the study by the readers.

-Results (Line 107-111): Breath isoprene has been suggested to be related to age (doi: 10.1515/CCLM.2008.181). Thus, adding data on the relationship between breath isoprene and age (e.g., correlations) in this study as supplemental data would be helpful in expanding the knowledge in this field.

-Results (Line 151-159): Did you find those who had IDI2 variants but were found with normal isoprene levels? Also, did you find IDI2 variants in the isoprene-deficient group (< 50ppV)?

-Discussion (Line 200-201): Please provide some reason rationale or references for setting < 50 ppbV as the cutoff and 100 – 300 ppbV as the normal range.

-Discussion (Line 213-217): Adding a diagram of the assumed human isoprene production pathway with IDI2 from the results of this study may help readers understand. A simple one would be enough.

-Discussion (Line 274-278): This study did not measure breath isoprene levels during exercise. Thus, you should indicate that an investigation of the relationship between breath isoprene during exercise and IDI2 is needed in the future.

-Figure 1: Are there differences between young (> 20 years) and senior (> 60 years) groups?

-Figure 1: Are there differences between the absent group and total inspiratory samples? If yes, you should indicate the reason in the discussion section.

-Table S1: Please state the information of statistical test for using the table in the legend.

Reviewer #3 (Remarks to the Author):

It was a great joy reading this manuscript by Dr Sukul and co-authors describing how they traced the molecular genetic origins of breath isoprene in humans. The exome sequencing, filtering, and analysis that isolated the IDI2 rs1044261 variant (p.Trp144Stop) as causative of isoprene absence was very well done.

I note that this particular IDI2 rs1044261 variant has a moderately high frequency in the general population (approximately 7 percent allele frequency in European ancestry participants, according to gnomAD). This frequency is high enough (especially in terms of heterozygous carriage) to explain a

large proportion of isoprene deficiency in the population. In this light, i have two humble suggestions for the Authors to consider, with the view of improving the manuscript to interest the readership of Communications Biology.

1. Can the Authors clarify whether all 2000 enrolled participants (line 96, main text) were unrelated to one another (not first or second degree relatives)?

2. Of the 64 participants who were clearly isoprene deficient (line 97, main text), how many carried one copy of the IDI2 rs1044261 variant? I humbly think that addressing this question will be extremely helpful to the readers.

Origin of breath isoprene in humans is revealed via multi-omic investigations

Pritam Sukul^{1*}†, Anna Richter²†, Christian Junghanss², Jochen K Schubert¹, Wolfram Miekisch¹

¹Rostock Medical Breath Research Analytics and Technologies (ROMBAT), Dept. of Anesthesiology, Intensive Medicine and Pain Therapy, University Medicine Rostock, Schillingallee 35, 18057 Rostock, Germany.

²Department of Medicine, Clinic III – Hematology, Oncology, Palliative Medicine, Rostock University Medical Center, Ernst-Heydemann-Strasse 6, 18057 Rostock, Germany.

*e-mail: pritam.sukul@uni-rostock.de

† These authors contributed equally to this work.

Via this letter we have provided ‘point-by-point’ response to all comments from the peer-reviewers. In our humble opinion, the expert comment and suggestions have significantly improved our present manuscript. We are grateful for their valuable time, constructive comments and efforts to enhance the clarity and readability of our presentation.

Reviewer comments are written in black.

Answers to the reviewers are written in blue.

Changes in the text are **highlighted** (Corresponding page and paragraph numbers refer to the revised manuscript). Corresponding line numbers are referred in all applicable responses.

Reviewer #1 (Remarks to the Author):

Review of the article

Origin of breath isoprene in humans is revealed via multi-omic investigations

Article authors: Pritam Sukul, Anna Richter, Christian Junghanss, Jochen K. Schubert, Wolfram Miekisch

Isoprene was first detected in breath in 1960 by gas chromatography equipped with an ionization detector [1]. In 1969, Jansson and Larsson [9] reported the first gas chromatography/mass spectrometry quantitative identification of isoprene in breath, giving a range of 0.09 – 0.45 ppm. Then in 1981, using 30 volunteers, with the exception of one individual, Gelmont et al. identified isoprene as the main endogenous hydrocarbon of human breath, the amount ranging from 30 to 70% of the total hydrocarbons exhaled.

Early discussions on isoprene production suggested that it derives from cholesterol biosynthesis via a mevalonate pathway, mainly by the liver. Certainly, it is evident from the studies of Deneris et al. [4] that isoprene is produced non-enzymatically by acid-catalyzed formation from dimethylallyl pyrophosphate [5]. This led to

suggestions that the exhaled breath concentrations of isoprene could be indirectly used to provide a measure of body cholesterol. In 1993, Stone et al. [18] showed that lovastatin, a competitive inhibitor of the rate-limiting step of cholesterol biosynthesis, significantly reduced breath isoprene levels. However, statins are known to have muscular side effects, the mechanism of which is unclear [2]. In addition, it is now well-proven that there is no correlation between breath isoprene concentrations levels and blood cholesterol levels [21, 14, 13, 19]. Thus, the major production of isoprene in the body must be from a different source.

Isoprene is low-soluble in blood as reflected by a small blood:gas partition coefficient $\lambda_{b:air} \approx 1.0$ [16] and hence the end-tidal breath concentration is given by the alveolar concentration. A simple mass balance model yields the Farhi equation [6] which relates the mixed venous concentration C_v with the alveolar concentration C_A by $C_A = C_v / (\lambda_{b:air} + V' / Q')$. Here V' / Q' is the ventilation-perfusion ratio which is approximately 1 at rest. This model easily explains the high sensitivity of the breath concentration on changes of blood flow and breath flow (breath maneuvers like hyperventilation or posture).

However, in contrast to the prediction by the Farhi equation breath isoprene concentration exhibit instead of a decrease a significant peak at the beginning of exercise on an ergometer [11]. King et al. [12] showed by mathematical modeling with mass balance differential equations that real-time breath isoprene concentration profiles during exercise could only be explained by the assumption that the main production of endogenous isoprene occurs in muscle tissue. Proved by an ergometer exercise with leg change. Already Miekisch et al. [15] investigated blood samples from selected vascular compartments of 19 mechanically ventilated pigs and discovered that isoprene was not equally distributed among the vascular compartments. Isoprene tended to be higher in portal and mixed venous blood and lower in hepatic venous and in arterial blood. Miekisch et al. concluded that the decreasing isoprene concentrations in the hepatic venous blood resulted from isoprene being metabolized in the liver and that the high mixed venous concentrations suggested that there is a peripheral origin of isoprene, e.g., muscle cells.

The 2021 Sukul et al. [19] study as well as that by King et al. [13, 22] provided further evidence that breath isoprene is not correlated with cholesterol levels. Sukul et al. demonstrated that endogenous isoprene does not originate from cholesterol synthesis by quantitative gene expression analysis of the mevalonate pathway enzymes of the female volunteer without isoprene in breath.

Although isoprene is a commonly found endogenous volatile in exhaled breath in high levels, as mentioned previously a few studies have found subjects (approximately 1 in a 1000) without isoprene in breath [7, 4, 20, 21, 14, 19, 3, 8]. The cause of this lack of production was unknown until now. Mochalski et al. [17] summarized the evidence for the origin of endogenous isoprene in muscle tissue in chapter 5 and suggested that "Investigations involving people who have no measurable isoprene in their exhaled breath can be expected to provide

an interesting insight into the possible production pathways of isoprene.”

The article discussed here did just that and provides definitive proof that endogenous isoprene originates from muscular lipolytic cholesterol metabolism.

The article is well-written, all measurements were carried out carefully, and the reported results seem correct and are in good agreement with other reports. I recommend it for publication with minimal changes.

Thank you for re-emphasizing the importance and the historical background of exhaled isoprene in human biology, biomarker research and clinical science. Despite several clues (Miekisch *et al* 2001, Turner *et al* 2005, and King *et al* 2010 – among others) towards a probable extrahepatic origin (e.g., from the muscles), any down-stream evidence to disqualify the putative origin was lacking until 2021. In line with our 2021 report (Sukul *et al.* 2021 [19]) that has finally disqualified the putative human isoprene origin, the recent review of this research area by Mochalski *et al.* 2023 [17] has re-summarized the available evidences and reemphasized our current research direction involving genetic analysis in rare adults.

After pioneering the recessive inheritance of the rare character in 2021 we continued searching for rare adults in consecutive clinical screening studies and simultaneously started the described multi-omic investigations in 2021 to finally unravel the origin of human exhaled isoprene from muscular metabolism.

We concur that the present finding provided definitive proof of the previous clues regarding the potential extrahepatic / muscular origin. We have added a brief overview within the following new section (Line 59) of the **Introduction**:

While looking at the physio-metabolic aspects, the increase in isoprene exhalation at the beginning of exercise was first reported in 1997³². In 2001, Miekisch *et al* first postulated the peripheral source of isoprene by measuring blood isoprene from vascular compartments of mechanically ventilated pigs³³. In 2005 Turner *et al* found one isoprene deficient healthy adult³⁴. While compared to isoprene normal adults, no correlations were seen between isoprene exhalation and fasting blood cholesterol profiles. In 2010, King *et al* also predicted its extrahepatic endogenous production via physiological modeling of exhaled isoprene dynamics³⁵. The model reasonably explained the exercise driven immediate increase in breath isoprene. In 2012, King *et al* observed reduced levels (by a factor ≥ 8 , compared to healthy adults) of blood and breath isoprene in five late state muscle dystrophy patients and thereby, postulated its possible production in muscle²⁹. In 2015, Unterkofler *et al* applied a two compartmental model to establish connection between endogenous production and metabolism of systemic VOCs and demonstrated that inhaled deuterated isoprene-D5 does not exhibit a peak at the beginning of exercise³⁶. Although, we have reported pronounced effects of peripheral vasoconstrictions (in muscular compartments) during standing³⁷, while wearing medical face-masks³⁸ and also effects of natural menstrual cycle and daily oral contraceptive pills³⁹ on exhaled isoprene profiles, actual down-stream analysis to confirm the true origin in human and metabolic source of breath isoprene was completely missing. In 2021, we executed breathomics, lipid profiling and gene expression analyses in an isoprene absent rare German adult and her isoprene deficient parents and sibling sister. Outcomes depicted no aberration in cholesterol levels and/or in gene expression of the mevalonate pathway enzymes and indicated a recessive inheritance of this healthy character³⁰. Therefore, we questioned the putative human origin (hepatic cholesterologenesis) of exhaled isoprene that was proposed (in 1984 by Deneris *et al*) based on *in vitro* experiments in rat liver cells⁴⁰. Nevertheless, a single rare case was insufficient for detailed down-stream multi-omic analysis to determine the exact source.

In a recent case study on one isoprene deficient American adult male and his blood-relatives by Harshman *et al*⁴¹ and in another study on an isoprene deficient Italian adult female and her blood-relatives by Biagini *et al*⁴² also demonstrated no relation of exhaled isoprene profiles to plasma cholesterol levels. They neither find any isoprene absent rare adult nor investigated the human cholesterol metabolism related gene expressions.

Minor required changes:

line 176: The following sentence is not correct because it is too general, and the sentence should therefore be modified as below.

"For the last 39 years, we have erroneously regarded hepatic cholesterologenesis (producing > 90% of human cholesterol) as the prime origin of human exhaled isoprene."

"For the last 39 years, many have erroneously regarded hepatic cholesterologenesis (producing > 90% of human cholesterol) as the prime origin of human exhaled isoprene. However, an article by King *et al.* [12]) provided evidence that this assumption is incorrect."

As per reviewer's suggestion the statement (Line 180) is now modified as the following:

For the last 39 years, many have erroneously regarded hepatic cholesterologenesis (producing > 90% of human cholesterol) as the prime origin of human exhaled isoprene.

We have also added the following statements (Line 186):

However, reports by Miekisch *et al*³³, Turner *et al*³⁴ and King *et al*³⁵ indicated extrahepatic production of this VOC and physiological modeling approaches under exercise hinted toward a probable muscular origin^{29,36}. Despite a few studies reported the rare presence of isoprene absent adults^{25,31,34,40-43}, until 2021, there was no down-stream evidence available to completely disregard the convention.

In addition, for completeness, at least the following articles should be cited, and the details should be provided, as in the text in brackets:

As per reviewer's suggestions, we have now added the following references and provided corresponding details within the introduction as per their scientific and factual appropriateness.

[1], (first report on isoprene in the breath)

Change to manuscript (Line 39):

In 1960, mass-spectrometric techniques had detected the presence of exhaled isoprene, which was later quantified in 1969^{11,12}.

[7], (first report of a person without isoprene in the breath)

Change to manuscript (Line 53):

Existence of an isoprene absent adult was first reported in 1981³¹. Since then, only a few pilot studies have randomly reported the presence of breath isoprene absent healthy subjects and in 2021, we have first approximated the actual rare genetic occurrence of this character by screening in a large cohort of 1026 humans³⁰.

[10] (first report on the increase of isoprene in the breath at the beginning of exercise)

Change to manuscript (Line 59):

While looking at the physio-metabolic aspects, the increase in isoprene exhalation at the beginning of exercise was first reported in 1997³⁶.

[12] (the first and only model that predicted that endogenous isoprene is produced in muscle tissue. This is the reason for the increase of isoprene in the breath at the beginning of exercise. Proved by an ergometer exercise with leg change.)

Change to manuscript (Line 64):

In 2010, King *et al* also predicted its extrahepatic endogenous production via physiological modeling of exhaled isoprene dynamics³⁷. The model reasonably explained the exercise driven immediate increase in breath isoprene assuming a muscular origin of the substance.

[22] (demonstrated that inhaled isoprene-d5 does not exhibit a peak at the beginning of exercise.)

Change to manuscript (Line 69):

In 2015, Unterkofler *et al* applied a two compartmental model to establish connection between endogenous production and metabolism of systemic VOCs and demonstrated that inhaled deuterated isoprene-D5 does not exhibit a peak at the beginning of exercise³⁹.

REFERENCES

1. Anonymous, SwRI pegs trace compounds in body fluids, Chemical & Engineering News Archive 38 (1960), no. 11, 36-37.
2. Jerzy Beltowski, Grazyna Wojcicka, and Anna Jamroz-Wisniewska, Adverse effects of statins - mechanisms and consequences, Current Drug Safety 4 (2009), 209-228.
3. Denise Biagini, Jonathan Fusi, Annasilvia Vezzosi, Paolo Oliveri, Silvia Ghimenti, Alessio Lenzi, Pietro Salvo, Simona Daniele, Giorgia Scarfo, Federico Vivaldi, Andrea Bonini, Claudia Martini, Ferdinando Franzoni, Fabio Di Francesco, and Tommaso Lomonaco, Effects of long-term vegan diet on breath composition, Journal of Breath Research 16 (2022), no. 2, 026004.
4. E. S. Deneris, R. A. Stein, and J. F. Mead, In vitro biosynthesis of isoprene from mevalonate utilizing a rat liver cytosolic fraction, Biochem. Biophys. Res. Commun. 123 (1984), 691-6.
5. _____, Acid-catalyzed formation of isoprene from a mevalonate-derived product using a rat liver cytosolic fraction, J. Biol. Chem. 260 (1985), no. 3, 1382-5.
6. L. E. Farhi, Elimination of inert gas by the lung, Respiration physiology 3 (1967), no. 1, 1-11.
7. D. Belmont, R. A. Stein, and J. F. Mead, Isoprene-the main hydrocarbon in human breath, Biochem. Biophys. Res. Commun. 99 (1981), 1456-1460.

8. Sean W Harshman, Anne E Jung, Kraig E Strayer, Bryan L Alfred, John Mattamana, Alena R Veigl, Aubrianne I Dash, Charles E Salter, Madison A Stoner-Dixon, John T Kelly, Christina N Davidson, Rhonda L Pitsch, and Jennifer A Martin, Investigation of an individual with background levels of exhaled isoprene: a case study, *Journal of Breath Research* 17 (2023), no. 2, 027101.
9. B. O. JANSSON and B. T. LARSSON, Analysis of organic compounds in human breath by gas chromatography-mass spectrometry, *J. Lab. & Clin. Med* 74 (1969), no. 6, 961-966.
10. Alfons JORDAN, Armin HANSEL, Carsten WARNECKE, Rupert HOLZINGER, Peter PRAZELLER, Wolfgang VOGEL, and Werner LINDINGER, On-line trace gas analysis at ppt-levels: Medical applications, food research and air quality, *Ber. nat.-med. Verein Innsbruck* 84 (1997), 7-17.
11. T Karl, P Prazeller, D Mayr, A Jordan, J Rieder, R Fall, and W Lindinger, Human breath isoprene and its relation to blood cholesterol levels: new measurements and modeling, *J Appl Physiol* 91 (2001), 762-70.
12. J. King, H. Koc, K. Unterkofler, P. Mochalski, A. Kupferthaler, G. Teschl, S. Teschl, H. Hinterhuber, and A. Amann, Physiological modeling of isoprene dynamics in exhaled breath, *J. Theor. Biol.* 267 (2010), 626-37.
13. J. King, P. Mochalski, K. Unterkofler, G. Teschl, M. Klieber, M. Stein, A. Amann, and M. Baumann, Breath isoprene: muscle dystrophy patients support the concept of a pool of isoprene in the periphery of the human body, *Biochem. Biophys. Res. Commun.* 423 (2012), 526-530.
14. I. Kushch, B. Arendacka, S. Stolc, P. Mochalski, W. Filipiak, K. Schwarz, L. Schwentner, A. Schmid, A. Dzien, M. Lechleitner, V. Witkovsky, W. Miekisch, J. Schubert, K. Unterkofler, and A. Amann, Breath isoprene-aspects of normal physiology related to age, gender and cholesterol profile as determined in a proton transfer reaction mass spectrometry study, *Clin. Chem. Lab. Med.* 46 (2008), 1011-1018.
15. W. Miekisch, J. K. Schubert, D. A. Vagts, and K. Geiger, Analysis of volatile disease markers in blood, *Clin. Chem.* 47 (2001), 1053-1060.
16. P. Mochalski, J. King, A. Kupferthaler, K. Unterkofler, H. Hinterhuber, and A. Amann, Measurement of isoprene solubility in water, human blood and plasma by multiple headspace extraction gas chromatography coupled with solid phase microextraction, *J. Breath Res.* 5 (2011), no. 4, 046010.
17. P Mochalski, J King, C A Mayhew, and K Unterkofler, A review on isoprene in human breath, *Journal of Breath Research* 17 (2023), no. 3, 037101.
18. B. G. Stone, T. J. Besse, W. C. Duane, C. D. Evans, and E. G. DeMaster, Effect of regulating cholesterol biosynthesis on breath isoprene excretion in men, *Lipids* 28 (1993), 705-708.
19. Pritam Sukul, Anna Richter, Jochen K. Schubert, and Wolfram Miekisch, Deficiency and absence of endogenous isoprene in adults, disqualified its putative origin, *Heliyon* 7 (2021), no. 1, e05922.
20. J. Taucher, A. Hansel, A. Jordan, R. Fall, J. H. Futrell, and W. Lindinger, Detection of isoprene in expired air from human subjects using proton-transfer-reaction mass spectrometry, *Rapid Commun. Mass Spectrom.* 11 (1997), 1230-1234.
21. C. Turner, P. Spanel, and D. Smith, A longitudinal study of breath isoprene in healthy volunteers using selected ion flow tube mass spectrometry (SIFT-MS), *Physiol. Meas.* 27 (2006), 13-22.
22. K. Unterkofler, J King, P. Mochalski, M. Jandacka, H. Koc, S. Teschl, A. Amann, and G. Teschl, Modeling-based determination of physiological parameters of systemic VOCs by breath gas analysis: a pilot study, *J. Breath Res.* 9 (2015), no. 3, 036002.

Reviewer #2 (Remarks to the Author):

The manuscript submitted by Sukul et al. has examined the metabolic origin of endogenous isoprene in humans by multi-omic analysis. The submitted work demonstrates that IDI2 plays a crucial role in the determination of isoprene production in humans. In general, the study is carefully designed and presented clear and significant results. The present study has made an important step toward understanding the mechanisms underlying human isoprene production. Thus, I certainly feel that the paper is important to the field. Meanwhile, there are several relatively minor concerns with the manuscript as written that must be addressed to help precise comprehension of the study by the readers.

We are delighted to see that the reviewer has clearly emphasised on the novelty, importance of our findings and have recognised the scientific rigour of our manuscript. All concerns raised by the reviewer are certainly important and we have tried our best to address all of those point-by-point.

-Results (Line 107-111): Breath isoprene has been suggested to be related to age (doi: 10.1515/CCLM.2008.181). Thus, adding data on the relationship between breath isoprene and age (e.g., correlations) in this study as supplemental data would be helpful in expanding the knowledge in this field.

Thank you. In our recent breathomics study (Sukul *et al.* 2022, *iScience*, **25**, 103739) on 204 healthy women aged between 5 – 85 years, we had executed a correlation analysis between age and exhaled breath isoprene. A relatively weak (R-value: 0.328) but statistically significant (p-value: <0.001) correlation was observed. As per reviewer’s suggestion we have now applied the correlation analysis and the results are as the following:

Age vs. Breath isoprene	Correlation coefficient
Overall (n = 2000)	0.045
All without isoprene aberrations (n = 1931)	0.057
Age <20 years (n = 345)	0.506
Age 20 - 60 years (Isoprene normal, n = 1318)	0.113
Age 20 - 60 years (Isoprene deficient, n = 64)	- 0.208
Age >60 years (n = 268)	- 0.433

Table S1. Correlations between biological age (year) and exhaled isoprene concentrations (ppbV).

As we have added the above table as ‘Table S1’, the order (/numbers) of existing supplementary tables changed accordingly. Table S1 is also referred to the **Results** section (Line 103) and the following statement is added to **Methods** section (Line 452):

Overall and group-wise correlations between subject’s biological age (year) and exhaled isoprene concentrations (ppbV) were assed via Spearman correlation test (due to non-parametric distribution of data).

-Results (Line 151-159): Did you find those who had *IDI2* variants but were found with normal isoprene levels? Also, did you find *IDI2* variants in the isoprene-deficient group (< 50ppV)?

Within the present study, homozygous presence of the *IDI2* variant was found via whole exome sequencing in the isoprene absent rare adults and was checked via targeted Sanger sequencing in available blood related adult family members of the rare adult-1 along with in two other isoprene normal healthy adults as controls. Nevertheless, due to resource constrain we could not check for the *IDI2* variant in all subjects.

We believe that screening of large population for *IDI2* may offer additional information but such approach is considerably expensive in a large scale and remained beyond the present study scope and our current financial capabilities. Therefore, we have now added the new statement within the following **Discussion** section (Line 217):

A broader approach of parallel *IDI2* sequencing in combination with breath analytics, however, is necessary to definitely confirm the relation between the amount of exhaled isoprene and *IDI2* genotype. Nevertheless, such approach is considerably resource demanding (in a large scale) and remained beyond our immediate aim and present resource capacity.

Yes, we have found heterozygous presence of the *IDI2* variant in all isoprene deficient subjects (both parents and sibling sister of the isoprene absent adult-1) who were actually checked for the mutation via Sanger sequencing. Please refer to the following statements within the **Results** section:

“Exhaled alveolar isoprene concentrations (corresponding room air subtracted) in rare adult-1’s father (aged 60 years, German), mother (aged 60 years, German) and sibling sister (aged 30 years, German) were 15.86 ppbV, 17.54 ppbV and 27.24 ppbV, respectively. These three adults were isoprene deficient and healthy.”

“Family members of isoprene absent adult-1, who demonstrated <30 ppbV of isoprene in exhaled breath, were also checked, revealing a heterozygous *IDI2* genotype in her mother, father and sibling sister.”

In order to address a multi-ethnic validation of the *IDI2* variant, we have planned a multi-country (Germany, Italy, Austria, Czech Republic and USA) study with colleagues who have also found subjects with isoprene deficiency or absence.

-Discussion (Line 200-201): Please provide some reason rationale or references for setting < 50 ppbV as the cutoff and 100 – 300 ppbV as the normal range.

The generally regarded normal range of 80 – 350 ppbV for exhaled isoprene in healthy adults is actually based on globally reported breath isoprene data from various physio-metabolic and clinical studies (including our own). To date, there is no true cut-off available for isoprene deficiency. Besides, exhaled isoprene concentrations in healthy adults may vary up to 25 ppbV due to normal/daily physiological (hemodynamics, ventilation and distribution etc.) and/or metabolic (e.g. natural menstrual cycle, oral contraception and menopause etc.) activities. As the majority of the presented isoprene data from our clinical screening studies are based on single-point measurements, we had set <50 ppbV to define a clear cut-off for isoprene deficiency i.e., beyond his/her physio-metabolic fluctuations.

In order to address reviewer’s concern, we had already presented the following rationales within the **Discussion** section of the manuscript (Line 221 and Line 226):

Besides, the cut-off limit of isoprene deficiency in adults was set by us to < 50 ppbV to define a clear cutoff beyond the inter-individual and physiological variations (normal range of exhaled isoprene 80 – 350 ppbV). Exhaled isoprene concentrations may vary by 5 – 25 ppbV in an individual simply due to his/her normal physiological variations in respiratory and hemodynamic parameters, natural menstrual rhythms and/or oral contraception and menopause^{23,40,42,43} etc. Therefore, the cut-off <50 ppbV allowed us to determine the clearly isoprene deficient adults – beyond his/her physio-metabolic fluctuations.

-Discussion (Line 213-217): Adding a diagram of the assumed human isoprene production pathway with *IDI2* from the results of this study may help readers understand. A simple one would be enough.

As per reviewer’s suggestion, we have provided the following diagram (as **Figure 5**) to address the assumed human isoprene production pathway with *IDI2*:

Figure 5: Isoprene production pathway with *IDI2* activity in human skeletal-myocyte. The pathway is based on the results of this study. The pathway depicts the potential contributions of myocellular organelles e.g., sarcoplasm, sarcoplasmic reticulum (SR), mitochondria and most importantly the peroxisomes. Human *IDI2* is only and highly expressed within the myocellular peroxisomes. Peroxisomes are metabolic organelle – mainly responsible for lipolysis. Here, acetyl-CoA is produced via beta-oxidation of fatty acids within peroxisomes. Acetyl-CoA is then channeled towards farnesyl-PP (FPP) production. One step before the FPP generation, isopentenyl-PP (IPP) is converted into its active isoform dimethylallyl-PP (DMAPP) by the *IDI2* activity. *IDI2* does not convert DMAPP back to IPP. In human, DMAPP is the only source of isoprene. Unlike plants, humans lack *isoprene synthase* and its enzyme homologue. Thus, *IDI2* determines human isoprene production in skeletal-myocytes. Farnesyl-PP exits peroxisomes and enters other organelles to execute in other metabolic pathways e.g., cholesterol metabolism in the SR.

The figure is now referred to the **Discussion** section (Line 312).

-Discussion (Line 274–278): This study did not measure breath isoprene levels during exercise. Thus, you should indicate that an investigation of the relationship between breath isoprene during exercise and *IDI2* is needed in the future.

Investigation of relationship between *IDI2* expression and isoprene exhalation under exercise was out of the present scope of the study. Previously we had investigated isoprene exhalations under exhaustive exercise to determine anaerobic threshold (Schubert *et al*, 2012, *Metabolomics*, **8**, 1069–1080) and ventilatory threshold (Pugliese *et al*, 2022, *Frontiers in Physiology*, **13**). As per our previous observations, change in isoprene exhalation under exercise depended on increased muscle activity along with perfusion, ventilation, compartmental distribution and washout phenomena.

Even if the “acute changes in *IDI2* gene expression” would need at least a few hours to be translated into the *IDI2* protein and fulfil its function, we will try to execute this interesting idea on a healthy cohort of adults in an upcoming study with the Institute of Sports Science (University of Rostock).

We have now added the following statement within the **Discussion** section (Line 308):

Investigation of relationship between *IDI2* gene and protein expressions and isoprene exhalation under exercise was beyond the scope of the present study design but may reveal interesting insights in future studies.

-Figure 1: Are there differences between young (> 20 years) and senior (> 60 years) groups?

We have realized that the reviewer actually wanted to mean young (< 20 years).

Yes, there are differences between young (< 20 years) and senior (> 60 years) groups. The statistical output is as the following:

SUMMARY

Groups	N	Average	Variance
Age <20 Years	345	86.506	3576.356
Age >60 Years	268	105.025	2469.624

ANOVA

Source of Variation	SS	df	MS	F	P-value	F crit
Between Groups	51724.886	1	51724.886	16.724	<0.001	3.856

We have now added the following statement within the **Results** section (Line 114):

The group of young subjects (aged <20 years) had significantly (p-value <0.001) lower isoprene levels than the group of seniors (aged >60 years).

-Figure 1: Are there differences between the absent group and total inspiratory samples? If yes, you should indicate the reason in the discussion section.

The isoprene absent group is different from any other group including the total inspiratory samples. This is due to the fact that the ambient air samples always represented very minimal but still detectable isoprene concentrations (median of 2.724 ppbV and 1.04 ppbV at 25% of the quartile and 4.63 ppbV at 75% of the quartile). In fact, this indicates the *in vivo* uptake/ metabolism and/or partial storage of inhaled isoprene fraction in isoprene absent subjects.

In order to comprehend the above facts, we have now modified and repositioned an existing statement within the **Discussion** section (Line 276) as the following:

The isoprene absent adults constantly inhaled low concentrations of ambient isoprene from the inspired air but they did not exhale any traceable fraction of this VOC. This indicates an uptake/ partial storage and/or metabolism of ambient isoprene in these individuals.

-Table S1: Please state the information of statistical test for using the table in the legend.

As per reviewer's suggestion, we have now added the statistical test description within the legend of **Table S2** (previously this was Table S1). Following changes are added to the supplement:

Table S2. Comparison of differences and statistical significance in isoprene concentrations between groups. Statistical significances were tested by means of Kruskal-Wallis one-way ANOVA on ranks (Kolmogorov-Smirnov test for normality followed by the multiple comparisons via post-hoc Dunn's method at p-value ≤ 0.005). Statistically significant differences with respect to 'isoprene normal adults (Iso.Nor_Adult)' are presented with

corresponding difference of ranks. Detailed data on each group size (N), group median, difference in ranks and corresponding p-values are presented.

In line with above, we have also added the following elaboration within the **Methods** section (Line 441):

Here, groups with young subjects (aged <20 years, n = 345), isoprene normal adults (aged 20 – 60 years, n = 1318), isoprene deficient adults (aged 20 – 60 years, n = 64), isoprene absent adults (aged 20 – 60 years, n = 5), seniors (aged >60 years, n = 268) and total inspiratory samples (n = 2000) were compared.

Reviewer #3 (Remarks to the Author) :

It was a great joy reading this manuscript by Dr Sukul and co-authors describing how they traced the molecular genetic origins of breath isoprene in humans. The exome sequencing, filtering, and analysis that isolated the *IDI2* rs1044261 variant (p.Trp144Stop) as causative of isoprene absence was very well done.

It is our genuine pleasure to see the positive remarks of the reviewer and will try our best to address the important concerns raised by the reviewer in order to improve our manuscript further.

I note that this particular *IDI2* rs1044261 variant has a moderately high frequency in the general population (approximately 7 percent allele frequency in European ancestry participants, according to gnomAD). This frequency is high enough (especially in terms of heterozygous carriage) to explain a large proportion of isoprene deficiency in the population. In this light, I have two humble suggestions for the Authors to consider, with the view of improving the manuscript to interest the readership of *Communications Biology*.

We second reviewer's views that the EU-ethnic prevalence of the *IDI2* rs1044261 variant (p.Trp144Stop) is relatively high (Table 1) and therefore, one may expect around 6.95% of the EU population to exhibit isoprene deficiency. Nevertheless, we could not determine the true occurrence of breath isoprene deficiency in our cohort due to the aspects indicated within the following section (Line 221 and Line 226) of the **Discussion**:

Besides, the cut-off limit of isoprene deficiency in adults was set by us to < 50 ppbV to define a clear cutoff beyond the inter-individual and physiological variations (normal range of exhaled isoprene 80 – 350 ppbV). Exhaled isoprene concentrations can fluctuate by 5 – 25 ppbV in an individual simply due to his/her normal physiological variations in respiratory and hemodynamic parameters, natural menstrual rhythms and/or oral contraception and menopause^{23,40,42,43} etc. Therefore, the cut-off <50 ppbV allowed us to determine the clearly isoprene deficient adults – beyond his/her physio-metabolic fluctuations.

1. Can the Authors clarify whether all 2000 enrolled participants (line 96, main text) were unrelated to one another (not first or second degree relatives)?

All 2000 subjects were enrolled in consecutive screening studies. It is not surprising that there were certain number of participants who belonged to same family (first- or second-degree relatives). Nevertheless, due to ethical obligations, we were not allowed to disclose/use any non-

anonymous/identifiable information and/or look into any relationships between these 2000 participants during the recruitment in those studies. Therefore, we did not consider such information during present data analysis.

Within the present study, we were only allowed to formally contact the available adult family members of the isoprene absent adults by means of official letters of invitation (to ask for voluntary participation).

N.B: As per the best of our knowledge, the five isoprene absent adults are unrelated to each other. Only the parents, adult sibling sister of the isoprene absent rare adult-1 and her infant daughter (reported earlier in Sukul *et al.* 2021, *Heliyon*) were confirmed blood relatives of each other. No other first or second-degree relatives of these isoprene absent adults were involved within the present study population.

We have now added the following statement within the **Discussion** section (Line 203):

It is possible, that a certain number of subjects from our clinical studies were first- or second-degree relatives. As per ethical obligations, we were not allowed to use/disclose such non-anonymous/identifiable information and/or to look for relationships between participants during their recruitments in clinical screening studies. Therefore, we did not consider such information during data analysis in our present study. Within this study, we were allowed to contact (only via formal letters of invitation) the adult family members of the isoprene absent adults. At the best of our knowledge, the five isoprene absent adults were unrelated to each other. Except the presented family members of the isoprene absent adult-1, no other blood-relatives of the remaining rare adults were involved within this study.

2. Of the 64 participants who were clearly isoprene deficient (line 97, main text), how many carried one copy of the *IDI2* rs1044261 variant? I humbly think that addressing this question will be extremely helpful to the readers.

We believe that screening of large population for *IDI2* may offer additional information but such approach is considerably costly in a large scale and remained beyond the present study scope and our current financial capabilities. Therefore, we had clearly indicated the following statement within the **Discussion** (Line 216):

A broader approach of parallel *IDI2* sequencing in combination with breath analytics, however, is necessary to definitely confirm the relation between the amount of exhaled isoprene and *IDI2* genotype.

As there was no direct funding available for this study, we consumed the remaining overheads from several projects to execute the present experiments. Thus, due to financial constrain we had to restrict the sequencing approaches to minimum and we could not check for the *IDI2* variant in all 64 isoprene deficient subjects.

Therefore, we have now added the new statement within the following **Discussion** section (Line 217): A broader approach of parallel *IDI2* sequencing in combination with breath analytics, however, is necessary to definitely confirm the relation between the amount of exhaled isoprene and *IDI2* genotype. Nevertheless, such approach is considerably resource demanding (in a large scale) and remained beyond our immediate aim and present resource capacity.

Of note, we have found heterozygous presence of the *IDI2* variant in all isoprene deficient subjects (both parents and sibling sister of the isoprene absent adult-1) who were actually checked for the mutation via Sanger sequencing.

Please refer to the following statements within the **Results** section:

“Exhaled alveolar isoprene concentrations (corresponding room air subtracted) in rare adult-1’s father (aged 60 years, German), mother (aged 60 years, German) and sibling sister (aged 30 years, German) were 15.86 ppbV, 17.54 ppbV and 27.24 ppbV, respectively. These three adults were isoprene deficient and healthy.”

“Family members of isoprene absent adult-1, who demonstrated <30 ppbV of isoprene in exhaled breath, were also checked, revealing a heterozygous *IDI2* genotype in her mother, father and sibling sister.”

REVIEWERS' COMMENTS:

Reviewer #1 (Remarks to the Author):

The authors have taken up all my suggestions.
Therefore, the manuscript can be published in its present form.

Reviewer #2 (Remarks to the Author):

I feel that the authors have sincerely and adequately addressed the concerns I have raised except for only one point. Please show each p-value of correlations in Table S1.
Congratulations on your fantastic work!

Reviewer #3 (Remarks to the Author):

The Authors have made a good faith attempt to address my suggestions (with a view to increasing reading pleasure by providing more scientific information that is sure to arise because of the large effect conferred by a relatively common allele), but they were unable to do so due to limitations that they kindly documented in the response.

In this light, i think the manuscript can be considered for acceptance, but the readership of the journal will likely be left hanging as to the status of those who are heterozygous (because the effect of the genetic allele is so large!)

Manuscript Number: COMMSBIO-23-2567A

Origin of breath isoprene in humans is revealed via multi-omic investigations

Pritam Sukul^{1*}†, Anna Richter²†, Christian Junghanss², Jochen K Schubert¹, Wolfram Miekisch¹

¹Rostock Medical Breath Research Analytics and Technologies (ROMBAT), Dept. of Anesthesiology, Intensive Medicine and Pain Therapy, University Medicine Rostock, Schillingallee 35, 18057 Rostock, Germany.

²Department of Medicine, Clinic III – Hematology, Oncology, Palliative Medicine, Rostock University Medical Center, Ernst-Heydemann-Strasse 6, 18057 Rostock, Germany.

*e-mail: pritam.sukul@uni-rostock.de

† These authors contributed equally to this work.

Via this letter we have provided ‘point-by-point’ response to the remaining comments from the peer-reviewers. We are grateful for their valuable time, constructive comments and efforts to enhance the clarity and readability of our presentation.

Reviewer comments are written in black.

Answers to the reviewers are written in blue.

Changes in the text are highlighted (Corresponding page and paragraph numbers refer to the revised manuscript). Corresponding line numbers are referred in all applicable responses.

Reviewer #1 (Remarks to the Author):

The authors have taken up all my suggestions.
Therefore, the manuscript can be published in its present form.

We thank the reviewer for his/her constructive comments and suggestions.

Reviewer #2 (Remarks to the Author):

I feel that the authors have sincerely and adequately addressed the concerns I have raised except for only one point. Please show each p-value of correlations in Table S1.
Congratulations on your fantastic work!

We thank the reviewer for his/her constructive comments, suggestions and kind wishes.

As per your suggestion, we have now added the corresponding p-values to the **Supplementary Table 1**.

This is to mention that we have encountered a transcription/manual error within the ‘Supplementary Table 1’. The error was caused while the correlation coefficient matrix was copied from a wrong column of the data output. We have now corrected the table accordingly. The table legend is also elaborated to increase comprehension. The changes are as the following:

Age vs. Breath isoprene	Correlation coefficient	p-value
Overall (n = 2000)	0.134	<0.001
All without isoprene aberrations (n = 1931)	0.159	<0.001
Age <20 years (n = 345)	0.520	<0.001
Age 20 - 60 years (Isoprene normal, n = 1318)	0.069	0.012
Age 20 - 60 years (Isoprene deficient, n = 64)	- 0.135	0.286
Age >60 years (n = 268)	- 0.413	<0.001

Supplementary Table 1. Correlations between biological age (in years) and exhaled isoprene concentrations (ppbV). Spearman correlation test was applied due to the non-parametric (as per Shapiro-Wilk test for normality) distributions of data. Number of observations (n), correlation coefficient and corresponding p-value are provided.

Reviewer #3 (Remarks to the Author):

The Authors have made a good faith attempt to address my suggestions (with a view to increasing reading pleasure by providing more scientific information that is sure to arise because of the large effect conferred by a relatively common allele), but they were unable to do so due to limitations that they kindly documented in the response.

In this light, i think the manuscript can be considered for acceptance, but the readership of the journal will likely be left hanging as to the status of those who are heterozygous (because the effect of the genetic allele is so large!)

We thank the reviewer for his/her constructive comments and suggestions.

We also second reviewer's thought that the readership of *Communication Biology* may has to wait until our next communications regarding the *IDI2* heterozygosity in relation to the true upper-limit of isoprene deficiency in adults.